# Tailored Uniaxial Alignment of Nanowires Based on Off-Center Spin-Coating for Flexible and Transparent Field-Effect Transistors

**DOI:** 10.3390/nano12071116

**Published:** 2022-03-28

**Authors:** Giwon Lee, Haena Kim, Seon Baek Lee, Daegun Kim, Eunho Lee, Seong Kyu Lee, Seung Goo Lee

**Affiliations:** 1Department of Chemical Engineering, Pohang University of Science and Technology, Pohang 37673, Korea; giwonlee@postech.ac.kr (G.L.); haenak@postech.ac.kr (H.K.); sbree@postech.ac.kr (S.B.L.); daegunkim@postech.ac.kr (D.K.); sklee84@postech.ac.kr (S.K.L.); 2Department of Chemical Engineering, Kumoh National Institute of Technology, Gumi 39177, Korea; leeeh@kumoh.ac.kr; 3Department of Chemistry, University of Ulsan, Ulsan 44610, Korea

**Keywords:** nanowire, spin-coating, alignment, field-effect transistor, flexible electronics

## Abstract

The alignment of nanowires (NWs) has been actively pursued for the production of electrical devices with high-operating performances. Among the generally available alignment processes, spin-coating is the simplest and fastest method for uniformly patterning the NWs. During spinning, the morphology of the aligned NWs is sensitively influenced by the resultant external drag and inertial forces. Herein, the assembly of highly and uniaxially aligned silicon nanowires (Si NWs) is achieved by introducing an off-center spin-coating method in which the applied external forces are modulated by positioning the target substrate away from the center of rotation. In addition, various influencing factors, such as the type of solvent, the spin acceleration time, the distance between the substrate and the center of rotation, and the surface energy of the substrate, are adjusted in order to optimize the alignment of the NWs. Next, a field-effect transistor (FET) incorporating the highly aligned Si NWs exhibits a high effective mobility of up to 85.7 cm^2^ V^−1^ s^−1^, and an on-current of 0.58 µA. Finally, the single device is enlarged and developed in order to obtain an ultrathin and flexible Si NW FET array. The resulting device has the potential to be widely expanded into applications such as wearable electronics and robotic systems.

## 1. Introduction

Over the past few decades, new approaches towards the alignment of nanowires (NWs) have been extensively investigated for use in versatile electrical applications such as field-effect transistors (FETs), energy harvesters, electrodes, and sensors [1,2,3,4,5,6]. In contrast to the conventional random geometries, the aligned NWs enable the utilization of their anisotropic properties within the electrical devices. In particular, the aligned semiconductor NWs have attracted much attention for improving the device performance because of their low operation voltages, high carrier mobilities, mechanical flexibility, and quantum confinement effects [7,8,9,10,11,12,13,14,15,16]. Hence, to facilitate the use of NWs as a component of integrated circuits, it is essential to control the process of NW assembly along a preferred direction. In particular, the uniaxial alignment of NWs is essential for obtaining advanced properties in electronic devices.

To this end, numerous researchers have investigated alignment methods such as electric field-directed assembly [17,18,19], flow-assisted alignment [20,21], selective chemical patterning [22], Langmuir-Blodgett methods [23,24,25], blown-bubble film techniques [26,27], and contact printing [28,29]. However, these methods involve additional transfer processes on a target substrate, or complicated procedures such as electric fields, fluid-flow channels, chemical surface treatment, and bulky equipment. Alternatively, spin-coating is a simple and fast process that is widely used for fabricating uniform thin films, and it has been shown to produce aligned nano- or micro-structures such as oriented conjugated polymer thin films [30], self-assembled nanoparticles [31], self-sorted and aligned single-walled carbon nanotube (SWCNT) networks [32,33], and many types of NWs [34,35]. For example, Bao’s group reported a simple spin-coating process for aligning SWCNT networks along the radial direction [32,33]. However, the SWCNTs were not uniaxially aligned throughout the entire area of the device, and the effective factors for alignment were not characterized. More recently, Zhou et al. proposed the alignment of silver nanowires on the basis of off-center spin-coating, and provided a limited explanation of the alignment that focused on the nanowire diameters and lengths [34]. However, to the best of the present authors’ knowledge, a detailed analysis of the NW alignment mechanism via the off-center spin-coating process has not been previously reported.

In the present work, the off-center spin-coating method is demonstrated for the fast and simple assembly of uniaxially-aligned NWs. In this method, the substrate is positioned away from the center of rotation in order to modulate the critical combination of the drag force, which is due to fluid flow, and the inertial forces from the rotational motion for successful alignment. In particular, the viscosity of the solvent and the acceleration time of the spin-coating are shown to crucially affect the drag force during spin-coating, which thereby induces the alignment of the NWs. Meanwhile, the distance between the substrate and the center of rotation during spin-coating is also important for aligning the NWs through the modification of the centrifugal force. In addition, the proposed outstanding method for controlling the morphology of NWs is used to fabricate a FET with an enhanced current and charge mobility, which is due to the lower junction-point density along the electrodes from the source to the drain that is provided by the uniaxially-aligned NWs. Finally, an ultrathin flexible FET array is developed in order to validate the usefulness of this approach for wearable electronic and robotic applications.

## 2. Results and Discussion

For use in the present study, vertically-aligned silicon nanowires (Si NWs) were initially synthesized via a metal-assisted chemical etching technique and were characterized by scanning electron microscopy (SEM) and high-resolution transmission electron microscopy (HR-TEM) (Appendix A). The full details are provided in the Appendix A. These NWs were then separated and dispersed in various alcohol solvents, and were utilized for the spin-coating process, as is shown schematically in Figure 1. When the substrate was placed at the center of rotation, as in the conventional spin-coating process (Figure 1a(i)), the Si NWs were deposited randomly in the radial direction from the center, rather than being aligned along the preferred direction (inset). By contrast, when the substrate was positioned away from the center of rotation (Figure 1a(ii)), the Si NWs were aligned uniaxially over the entire surface area (inset). The various external forces that can be applied to the Si NWs during solvent drying in the off-center spin-coating are shown schematically in Figure 1b. Thus, depending on the position of the substrate, the NWs can experience a drag force in the parallel direction to the fluid motion that results from the net inertial force that is due to the non-uniform circular motion. In this respect, the conventional and off-center spin-coating processes differ in the size and direction of the resultant force (i.e., the combined drag and inertial effects). In particular, during the off-center spin-coating process, the Si NWs that are dispersed in the solution droplet are all affected by the external centrifugal force along the outward radial direction, as is shown in Inertial Force I of Figure 1b. In addition, during the initial (acceleration) stage of spin-coating, the angular velocity increases rapidly, Inertial Force II of Figure 1b is noticeably elevated due to tangential acceleration, and the NWs that are in partial contact with the substrate strongly experience both types of force, as is shown schematically in Figure 1c. This can cause the three-phase contact line (TCL) to move into the same direction as the net force, which thereby allows those NWs to become aligned along the direction that is defined by the inertial force because of both centripetal acceleration and tangential acceleration [4,36].

The main components of the drag force are the frictional drag and the pressure. The former originates from the shear force between the Si NW and the solvent, while the latter is due to the pressure that is applied to the NWs. In the present work, however, the Si NWs have a high aspect ratio of about 600, and the ratio of inertial forces to viscous forces in the solvent (i.e., the Reynolds number (Re)) is low (~10^2^, which indicates laminar flow); hence, the Si NWs are predominantly affected by the frictional drag [37]. The frictional drag force (*F_Df_*) is given by Equation (1):(1)$$F_{Df\approx }\frac{\pi R\mu }{C_{Df}}\left(v-u\right)$$
where *C_Df_* is the coefficient of the frictional drag; *R* is the radius of the NW; *μ* is the viscosity of the solvent; *v* is the linear velocity of the NWs; and *u* is the fluid velocity (see Appendix A for more details).

As is shown in Equation (1), the viscosity and the relative velocity are proportional to the drag force. In other words, the solvent viscosity increases the shear force between the solvent and the substrate, which thereby inhibits the solvent from moving in the direction of the substrate. In effect, this increases the relative velocity between the fluid and the nanowires that are in partial contact with the substrate.

To determine the key parameters that affect the alignment of the Si NWs, the viscosity of the solvent was modified by using various solvents, such as ethanol, isopropyl alcohol (IPA), n-butanol, and isobutanol. The NW alignment was then characterized according to the alignment angle (*θ*) (i.e., the tilt of the NWs away from the direction of the centrifugal force), and the degree of randomness (DoR) (i.e., the standard deviation of *θ*) and the results are presented in Figure 2a. Here, *θ* is seen to increase as the viscosity of the solvent increases (from ethanol to isobutanol), whereas the DoR decreases, which thus indicates an increase in the drag force, which is followed by more NWs becoming uniaxially aligned (Figure 2b). When low-viscosity solvents such as ethanol and IPA are used (1.1 and 2.0 cP, respectively), the solvent is quickly evaporated because of its low boiling point (78 and 83 °C, respectively) and, hence, the Si NWs are almost randomly oriented (top panel, Figure 2b). However, when n-butanol or isobutanol are used, the increased viscosities (3.0 and 4.0 cP, respectively) and the boiling points (118 and 108 °C, respectively) allow the Si NWs to align uniformly and uniaxially (bottom panel, Figure 2b). These results suggest that the viscosity has a much greater effect on the alignment of the NWs when the boiling point is sufficiently high to allow the TCL of the solution to change during the spinning process.

In addition, the effects of the acceleration time during the initial stages of spin-coating are revealed in Figure 2c,d. After drop-casting the solution onto the substrate, some of the suspended NWs will be in partial contact with the substrate. When the spinning is initiated at the target speed effectively, instantaneously (labeled as an acceleration time of 0 s in Figure 2d), the Si NWs that are in partial contact with the substrate rapidly move in synchrony with the substrate. Consequently, the solvent is affected by a high drag force and, hence, the increase in the velocity of the NWs is much faster than that of the solvent. This high shear force and the inertial force due to tangential acceleration promote the alignment of the NWs by increasing the alignment angle (*θ*) and by decreasing the degree of randomness (DoR). However, when a longer acceleration time is applied, the instantaneous acceleration of the NWs decreases, and, hence, the linear velocity of the NWs relative to the solvent is decreased during the initial stages. In this case, the Si NWs are subjected to a low shear force and a reduced drag force, which thus results in poor alignment (Figure 2d, bottom panel). Taken together, these results demonstrate that the alignment of the NWs can be optimized by using a high-viscosity high-boiling solvent and a near-instantaneous acceleration.

The effects of the off-center spin-coating at various distances from the center of rotation are demonstrated in Figure 3a,b. As is shown schematically in Figure 3a, the position of the substrate was varied from the center (0) to the edge (9 cm) of rotation, and the resulting alignment angles (°) and degrees of randomness (DoR) of the NWs are plotted in Figure 3b. According to Marchell et al. [38], the centrifugal force is given by Equation (2):(2)$$F_{c\;}=\left(\chi -1\right)m_{N}\Omega_{s}{}^{2}r,\;\chi =\frac{\rho_{f}}{\rho_{p}}$$
where *m_N_* is the mass of the NW; *Ω_S_* is the angular rotation of the substrate; and *r* is the distance between the substrate and the center of rotation. In the present work, *ρ_p_* can be replaced by *ρ_N_* (the density of a Si NW), and, because the density of the solvent is similar to that of the NWs, *χ* can be ignored. Hence, Equation (2) can be modified as Equation (3):(3)$$F_{c}=m_{N}\Omega_{s}{}^{2}r$$

As is shown in Equation (3), the centrifugal force is proportional to the distance (*r*) of the substrate from the center of rotation. Consequently, the centrifugal force increases, and both the NW alignment angle and the DoR decrease as the distance of the substrate from the center of rotation is increased, as is shown in Figure 3b.

In addition, the NW alignment is affected by the surface energy of the substrate. In the present work, the substrate has a high surface energy because of the cleaning with piranha solution. Consequently, the substrate could be fully wetted by the solution, which thus leads to high frictional forces between the substrate and the nanowires in the solution droplet. Thus, when the substrate rotates very rapidly, the solution experiences a high shear force and, hence, the NWs become uniaxially aligned [4,36]. By contrast, when the surface energy of the substrate is low, the same volume of solution will not fully cover the surface and, hence, the frictional force between the substrate and the NWs in the droplet is also low. In such cases, the low shear force can allow a droplet of the solution to slide off the substrate, and a random morphology of the NWs can be observed [39,40].

As is shown in Appendix A, the density of the NWs can be easily controlled by changing the number of repeated spin-coating processes while using the same concentration of the solution. Here, the density of the NWs is seen to increase gradually as the number of spin-coating processes is increased. Moreover, this is followed by an increase in the number of connecting junctions between the NWs. As is shown in Appendix A, this versatile method can be widely applied to other target materials such as single-walled carbon nanotubes (SWNTs), which thus suggests that any materials that have a large aspect ratio can be uniaxially aligned via this novel method.

Finally, to confirm the usefulness of the proposed alignment method, the effects of the NW DoR values on the performance of the fabricated Si NW-based field-effect transistors (FETs) are presented in Figure 3c. Here, the inset images reveal that a lower DoR value corresponds to a larger number of bridging NWs between the source and drain electrodes, and fewer junctions between each NW. Moreover, the device with well-bridged Si NWs and few NW junctions (low DoR) exhibits a high on-current because of the smooth movement of the charge carriers through the single crystal in the absence of any obstacles. By contrast, the devices with high DoR values exhibit low on-currents because of the barriers that are formed by the many junctions between the NWs. As is shown in Figure 3d, the Si NW-based FET with a DoR value of 4 exhibits an on/off ratio of ~10^5^, and a threshold voltage of 6.81 V at a *V*_DS_ of −1 V. The effective linear field-effect mobility (μ*_eff_*) of an FET is given by Equation (4):(4)$$\mu_{eff}=L/\left(W_{eff}\;\cdot \;C_{i}\;\cdot \;V_{DS}\right)\;\cdot \;g_{m}$$
where *L* is the channel length; *W_eff_* is the effective channel width; *C*_i_ is the capacitance per unit area of the gate dielectric; and *g*_m_ is the transconductance. Thus, for the device in Figure 3d, the calculated μ*_eff_* is 85.74 cm^2^ V^−1^ s^−1^.

To further demonstrate the practical application of the proposed fabrication process, ultrathin (~2 μm) FET arrays of the aligned NWs were fabricated by off-center spin-coating onto parylene-C, as is shown schematically in Figure 4a. This substrate was chosen because it has been used previously for its thin and flexible properties [41]. Prior to spin-coating, the substrate surface of parylene-C was modified via oxygen plasma treatment, as is shown in Figure 4b. This rendered the surface hydrophilic, as was revealed by a decrease in the water-droplet contact angle (CA) from 85° before the treatment, to 15° afterwards. The NWs were then successfully aligned on the –OH-modified parylene-C surface according to the optimized experimental conditions that were described in the previous section. Because of the flexible and transparent properties of the parylene-C substrate, the obtained thin-film transistor (TFT) array is highly transparent and deformable, and it can easily be attached to various curved surfaces, as is shown in Figure 4c. Moreover, the as-prepared device is highly transparent to visible light (>90%), as is shown in Figure 4d. Furthermore, the electromechanical stability of the as-fabricated device under bending stress was measured by attaching it to human skin, to vials, and to a glass Pasteur pipette, as is shown in Figure 4e, and the resulting charge mobilities were calculated from the transconductance changes shown in Appendix A. Thus, the flexible device exhibits a mobility (μ*_eff_*) of 38.6 cm^2^ V^−1^ s^−1^ on human skin (left-hand image, Figure 4e), while the mobility increases from 24.5 cm^2^ V^−1^ s^−1^, to 35.6 cm^2^ V^−1^ s^−1^, to 37.5 cm^2^ V^−1^ s^−1^ as the bending radius increases from 3 mm, to 7.5 mm, to 14 mm. We measured the electrical characteristics of the output curve of the flexible devices that were attached to the various surfaces (Appendix A). The on-current value of each device decreases as the bending radius becomes smaller. Moreover, the non-ohmic contact is shown at the output curve of the FET that was measured at the bending radius of 3 mm, which implies that the contact resistance of FETs increases as the bending radius becomes smaller. Nevertheless, the transfer curves in Figure 4f show little difference according to the degree of bending, which thus indicates proper device operation and high electromechanical stability. To compare with the attachment of the curved surface, we tested the flexible device at the flat surface, and measured the electrical characteristics of the transfer curve in dual-sweep (forward and backward), which showed little hysteresis (Appendix A). The gate leakage current of the device with a flat surface was also measured and it showed two-order-lower current levels. It seems a slightly high level for the gate leakage, while that of the flexible substrate shows higher than the flat silicon substrate because of the roughness of the flexible substrate. Lastly, we repeatedly tested the flexible device and measured the electrical characteristic of the transfer curve twice in order to confirm the bending stability. The electrical transfer curves of the first and second sweeps showed at the same level, which represents the good bending stability of the cycles (Appendix A). Although Zheng’s group reports low-voltage OTFT flexible devices with excellent bending stabilities by using optimized poly(acrylic acid) and poly(ethylene glycol) dielectrics [42], our proposed flexible FETs also showed good bending stability, which was due to the highly aligned Si NWs through the channel direction, which maintains the electrical channel path, even in the bending state.

## 3. Conclusions

Herein, the one-step assembly method of off-center spin-coating was successfully demonstrated for the direct alignment of NWs without any additional transfer process or treatment. The detailed alignment mechanism was investigated by analyzing the effect of the applied external forces upon the behavior of the NWs during spin-coating, which indicated that the NW alignment originates from the resultant vector of the directions of the drag and inertial forces. A comparison of the corresponding formulas for the drag and centrifugal forces revealed that the NW alignment is influenced by factors such as the viscosity of the solvent, the acceleration time for spinning, and the distance between the substrate and the center of rotation. Hence, the proposed method is not only simple and fast, but also versatile, as the distance of the substrate from the center of rotation can be easily adjusted. Moreover, the technique can be easily applied on various high-aspect-ratio active materials to develop flexible electronics by directly patterning the aligned NW assembly onto the active layer, as is demonstrated herein by the fabrication of a large-scale, ultrathin flexible FET with an array structure. When attached to human skin, the device was shown to maintain a high charge mobility and a high on-current during body movement, which was due to the introduction of aligned NWs as an active component. Moreover, the device was shown to operate properly when attached to a curved surface. These results provide a novel and broad platform for applications in wearable devices and robotic systems.

## Figures and Tables

**Figure 1 nanomaterials-12-01116-f001:**
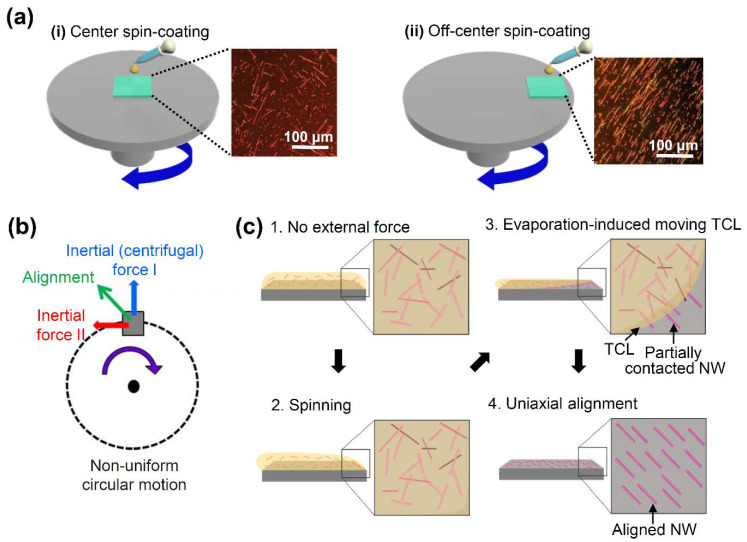
Schematic diagrams of the overall spin-coating processes and their effects upon NW alignment: (**a**) the conventional spin-coating setup (i) and the proposed off-center spin-coating setup (ii) with inset polarized optical microscope images of the as-deposited Si nanowires; (**b**) the forces involved in the off-center spin-coating mechanism, including Inertial (centrifugal) Force I, due to centripetal acceleration (blue), Inertial Force II (red), due to tangential acceleration, and the resultant force (green); (**c**) the sequential influence of the resultant force upon the uniaxial alignment of the NWs that are in partial contact with the substrate surface.

**Figure 2 nanomaterials-12-01116-f002:**
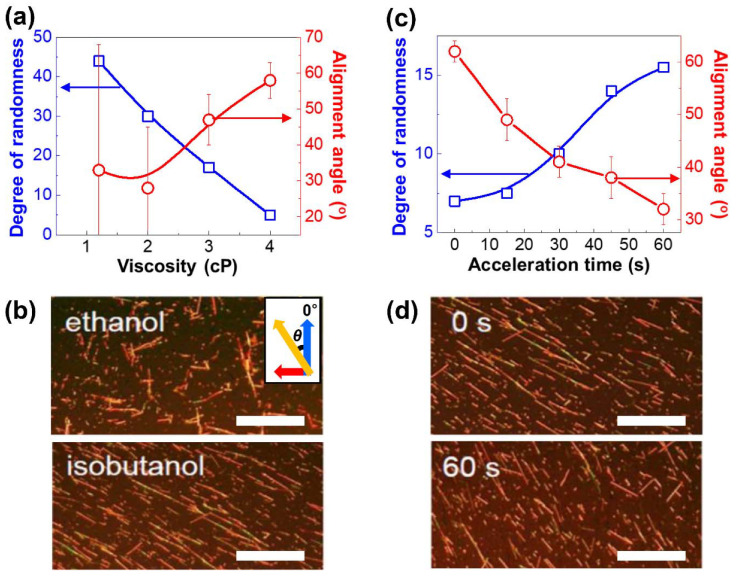
Affecting factors for NW alignment; (**a**) plot of the degree of randomness and alignment angle of the Si NWs as a function of viscosity; (**b**) polarized optical microscope (POM) images (scale bar = 100 μm) of the Si nanowire films spin-coated by using ethanol (**top** panel) and isobutanol (**bottom** panel), with the inset showing the alignment angle (*θ*) and the resultant force direction; (**c**) plot of the degree of randomness and alignment angle of Si NWs as a function of the acceleration time; (**d**) POM images (scale bar = 100 μm) of the Si nanowire films spin-coated by using acceleration times of 0 s (**top** panel) and 60 s (**bottom** panel).

**Figure 3 nanomaterials-12-01116-f003:**
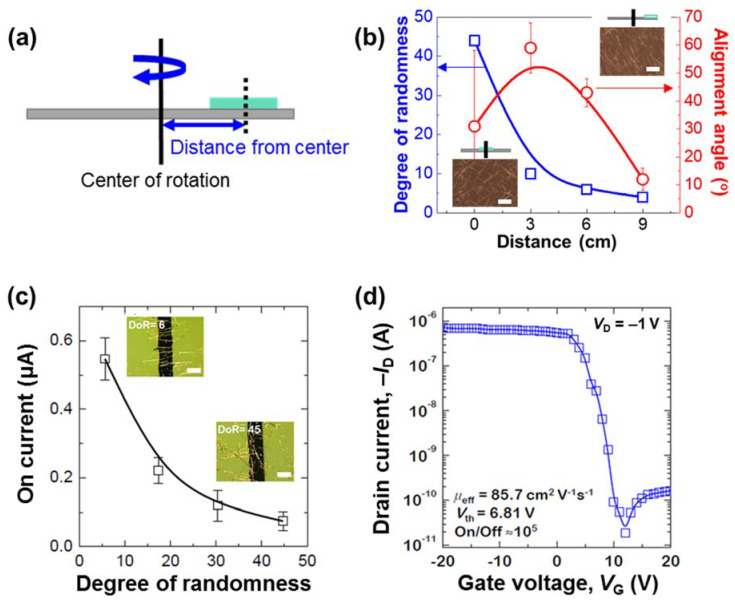
Correlation between NW alignment and FET device performance; (**a**) schematic diagram of the off-center spin-coating process with various distances of the substrate from the center of rotation; (**b**) plots of the degree of randomness (blue) and alignment angle (red) of Si NWs against various substrate distances (*r*), as defined in (**a**); (**c**) plot of the drain on-current at *V*_G_ = −20 V and *V*_D_ = −1 V against the degree of randomness of the Si NW films, with inset POM images (scale bar = 30 μm) of the Si NW-based FETs with DoR values of 6 and 45; (**d**) transfer characteristics of the FET devices (*V*_D_ = −1 V).

**Figure 4 nanomaterials-12-01116-f004:**
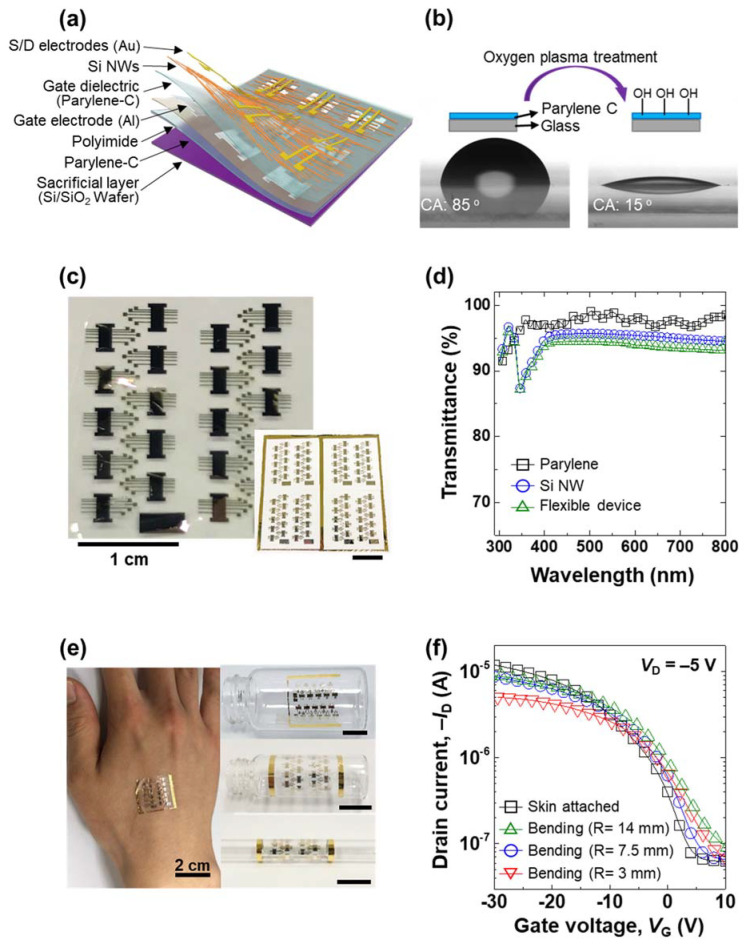
Applications for wearable and transparent devices; (**a**) schematic illustration of the flexible Si NW-based FET device; (**b**) schematic diagram and photographic images showing the effect of oxygen plasma treatment upon the water-droplet contact angle (CA) on the parylene-C film; (**c**) photographic images of the as-prepared flexible and transparent device; (**d**) transmittance curves of the parylene-C, the Si NW layer, and the integrated device; (**e**) flexible devices attached to human skin (**left** panel), and to glass vials and a glass Pasteur pipette (**right** panel); (**f**) transfer curves of the flexible device attached to the various surfaces (*V*_D_ = –5 V).

## Data Availability

Not applicable.

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
