# Peer review of "Tailored Uniaxial Alignment of Nanowires Based on Off-Center Spin-Coating for Flexible and Transparent Field-Effect Transistors"

_nanomaterials, 2022, doi:10.3390/nano12071116_

Round 1

Reviewer 1 Report

The authors present sound scientific reasoning for the role of spin parameters, and material/fluid parameters to describe the alignment of nanowires in a modified off-axis spin coat process. However, the practical utility of the approach for large area aligned nanostructures is not well addressed. How can the inherently uniform forces over large areas be used to cause parallel alignment. Also, how do inter-nanowire forces compete with those of the external forces to separate and straighten nanowires, and how do the considerations of surface adhesion compete with those of mobility of nanowires in solution. The authors should present results to explain these factors.

Author Response

Manuscript number: nanomaterials-1639301

MS-Type: Article

Title: " Tailored Uniaxial Alignment of Nanowires Based on Off-Center Spin-Coating for Flexible and Transparent Field-Effect Transistors "

Reply to Reviewer #1

Thank you for your invaluable comments.

We revised the manuscript according to your comments.

Comment 1:

How can the inherently uniform forces over large areas be used to cause parallel alignment?

Response:

We propose an off-center spin-coating method in which the external forces are controlled by positioning the target substrate away from the center of rotation. With this technique, various influencing factors such as the type of solvent, the spin acceleration time, the distance between the substrate and the center of rotation, and the surface energy of the substrate, are adjusted to optimize the alignment of the nanowires. These main factors consequentially develop the non-uniform forces within the nanowires over large areas during the off-center spin-coating. As a result, we can obtain the uniaxial alignment of nanowires in relatively small areas away from the center.

Comment 2:

How do inter-nanowire forces compete with those of the external forces to separate and straighten nanowires?

Response:

According to the previously published work, the forces between nanowires result from electrostatic interaction, mainly related to the distance between them (Nanomaterials, 2018, 8, 456). To reduce the inter-nanowire forces, we utilized an ultra-sonication technique for nanowires to be individually dispersed in the solution before using them for spin-coating. Also, a lower concentration of the nanowire solution was chosen to prevent the aggregation of nanowires. Therefore, we believe the forces between nanowires can be overcome during the coating process.

Comment 3:

How do the considerations of surface adhesion compete with those of mobility of nanowires in solution?

Response:

As we discussed in the manuscript, page 4, the substrate has a high surface energy due to the cleaning with piranha solution, leading to fully wetting of the solution and producing high frictional forces between the substrate and the nanowires in the solution droplet. In addition to the high frictional forces, the mobility of the nanowires in the solution allows more nanowires to contact the substrate. In this case, the site close to the surface of the substrate might not be experienced by the movement of the solution because the interface between substrate and solution could be fully contacted with a high friction force. For this reason, partially contacted nanowires on the substrate can be anchored even though the solution moves by external forces.

Reviewer 2 Report

Lee et al. presented the tailored uniaxial alignment of Si nanowires based on off-center spin-coating for flexible FETs. The results of this work may be useful to the design of semiconductor nanostructures and their FETs for wearable electronics and soft robotics, which qualify the scope of Nanomaterials. However, there are several problems which should be resolved before the publication.

  1. Where are output characteristics of those FETs? Please provide the related results and discussion.
  2. Please show the device performance diagram in which Si NWs are uniaxially oriented, but the direction is perpendicular to the channel direction.
  3. It is suggested to add the influence of spin-coating speed on the structure and performance of Si NWs-based films.
  4. Are there hysteresis phenomena in your transfer curves? Please provide the back-sweep data of the flexible FETs. At the same time, gate leakage currents of the devices also should be measured.
  5. The flexible Si NWs FETs showed good bending stability at different bending radii. Why? What about device performance of the inorganic Si NWs-based FET after long bending cycles? These should be made clear. Please also compare them with some impressive reports on polyelectrolyte dielectrics for flexible low-voltage organic transistors and wearable electronic devices with excellent bending stability.
  6. Did the experimenter agree with the human test? This statement should be added into the Experimental.

Author Response

Manuscript number: nanomaterials-1639301

MS-Type: Article

Title: " Tailored Uniaxial Alignment of Nanowires Based on Off-Center Spin-Coating for Flexible and Transparent Field-Effect Transistors "

Reply to Reviewer #2

Thank you for your invaluable comments.

We revised the manuscript according to your comments.

Comment 1:

Where are output characteristics of those FETs? Please provide the related results and discussion

Response:

We measured the electrical characteristics of the output curve of the flexible devices attached to the various surfaces (Figure S5 in Supporting Information). As the curvature of the attached surface decreases, the on-current value decreases. Moreover, we found that the contact resistance increases due to the increase of the on voltage. We added these statements in page 5.

Figure S5. The electrical characteristics of the output curve of the flexible devices attached to the various surfaces.

Comment 2:

Please show the device performance diagram in which Si NWs are uniaxially oriented, but the direction is perpendicular to the channel direction.

Response:

According to Figure 3c, we can expect that the case of aligned nanowires with perpendicular direction to the channel direction might possess the lowest on-current value due to the lack of the current path within the channel.

Comment 3:

It is suggested to add the influence of spin-coating speed on the structure and performance of Si NWs-based films.

Response:

As shown in manuscript, page 3, the frictional drag force is affected by the difference of the velocity between nanowire and fluid instead of the constant velocity of the spin-coating. With this knowledge, the acceleration time of spinning was controlled to observe the phenomenon of the alignment. Therefore, we did not perform a test with the influence of spin-coating speed.

Comment 4:

Are there hysteresis phenomena in your transfer curves? Please provide the back-sweep data of the flexible FETs. At the same time, gate leakage currents of the devices also should be measured.

Response:

We tested the flexible device attaching at the flat surface, and measured the electrical characteristic of transfer curve in dual-sweep (forward and backward), showing little hysteresis (Figure S6 in Supporting Information). We also measured the gate leakage current of the device with flat surface showing two orders lower current level. It seems a slightly higher level for the gate leakage, while that of the flexible substrate shows higher than the flat silicon substrate due to the roughness of flexible substrate.

Figure S6. The electrical characteristic of transfer curve of the flexible device with dual sweep (VDS = -5V).

Comment 5:

The flexible Si NWs FETs showed good bending stability at different bending radii. Why? What about device performance of the inorganic Si NWs-based FET after long bending cycles? These should be made clear. Please also compare them with some impressive reports on polyelectrolyte dielectrics for flexible low-voltage organic transistors and wearable electronic devices with excellent bending stability.

Response:

We repeatedly tested the flexible device and measured the electrical characteristic of the transfer curve twice to confirm the bending stability. The electrical transfer curves of the 1st and 2nd sweep showed at the same level representing good bending stability of cycles (Figure S6 in Supporting Information). Although Zheng’s group reported low-voltage OTFT flexible devices with excellent bending stability by using optimized poly(acrylic acid) and poly(ethylene glycol) dielectrics (Adv. Funct. Mater. 2019, 29, 1806092), our proposed flexible FETs also showed good bending stability due to the highly aligned Si NWs through the channel direction, which maintains the electrical channel path even in the bending state. This explanation is inserted in page 5.

Figure S6. The electrical characteristic of transfer curve of the flexible device with twice sweep.

Comment 6:

Did the experimenter agree with the human test? This statement should be added into the Experimental.

Response:

All procedures were approved by the Research Ethics Committee of Pohang University of Science and Technology in South Korea (PIRB-2020-E017). Written informed consents were obtained from all subjects. We added these statements in the Supporting information (page 3, line 15).

Round 2

Reviewer 1 Report

The revised manuscript addresses my previous concerns

Reviewer 2 Report

The author has done a good job of the revised manuscript.